# Early Cell Cultures from Prostate Cancer Tissue Express Tissue Specific Epithelial and Cancer Markers

**DOI:** 10.3390/ijms24032830

**Published:** 2023-02-01

**Authors:** Vladimir M. Ryabov, Mikhail M. Baryshev, Mikhail A. Voskresenskiy, Boris V. Popov

**Affiliations:** 1Department of Intracellular Signaling and Transport, Institute of Cytology, Russian Academy of Sciences, 194064 St. Petersburg, Russia; 2Institute of Microbiology and Virology, Riga Stradins University, Ratsupites 5, LV-1067 Riga, Latvia; 3City Multiprofile Hospital No. 2, Ministry of Health of the Russian Federation, 194354 St. Petersburg, Russia

**Keywords:** localized prostate cancer, prostate cell cultures, patient derived organoids (PDOs), prostate epithelial and cancer markers

## Abstract

Prostate cancer (PCa) is a widespread oncological disease that proceeds in the indolent form in most patients. However, in some cases, the indolent form can transform into aggressive metastatic incurable cancer. The most important task of PCa diagnostics is to search for early markers that can be used for predicting the transition of indolent cancer into its aggressive form. Currently, there are two effective preclinical models to study PCa pathogenesis: patients derived xenografts (PDXs) and patients derived organoids (PDOs). Both models have limitations that restrict their use in research. In this work, we investigated the ability of the primary 2D prostate cell cultures (PCCs) from PCa patients to express epithelial and cancer markers. Early PCCs were formed by epithelial cells that were progressively replaced with the fibroblast-like cells. Early PCCs contained tissue-specific stem cells that could grow in a 3D culture and form PDOs similar to those produced from the prostate tissue. Early PCCs and PDOs derived from the tissues of PCa patients expressed prostate basal and luminal epithelial markers, as well as cancer markers AMACR, TMPRSS2-ERG, and EZH2, the latter being a promising candidate to mark the transition from the indolent to aggressive PCa. We also identified various TMPRSS2-ERG fusion transcripts in PCCs and PDOs, including new chimeric variants resulting from the intra- and interchromosomal translocations. The results suggest that early PCCs derived from cancerous and normal prostate tissues sustain the phenotype of prostate cells and can be used as a preclinical model to study the pathogenesis of PCa.

## 1. Introduction

Prostate is a small auxiliary gland of the male reproductive system that is more susceptible to oncogenic transformation than other organs. One in eight men will be diagnosed with prostate cancer (PCa) during their lifetime. PCa is a major cause of cancer-related morbidity and mortality in developed countries (26% and 11%, respectively, in the USA in 2021) [1]. In many cases, PCa proceeds in the indolent form that does not require active therapy. However, it can transform into aggressive metastatic cancer that is resistant to medical treatment and leads to imminent death [2]. It was suggested that different PCa forms originate from different types of epithelial prostate cells and have distinct etiology and pathogenesis [3,4]. The molecular profiles of prostate epithelial cells and their ability to differentiate and reprogram into cancer cells are investigated in many leading biomedical laboratories worldwide [5,6,7]. One of the most important tasks in the PCa treatment is to search for the early markers of the dormant PCa transition into its aggressive form. There are two currently available preclinical models that effectively recapitulate PCa heterogeneity and are used to study PCa pathogenesis, development, markers, and treatment efficiency: patient derived xenografts (PDXs) and patient-derived organoids (PDOs) [8,9]. However, both models show some limitations. Namely, PDXs grow only in immunocompromised mice [10] and do not recapitulate all features of the original tumor because they are generated from a small amount of the tumor tissue [11]. Also, PDXs exhibit low growth rates, which increase over the course of passaging and depend on the PDX clonal selection [12]. PDOs originate from normal or cancer somatic stem cells (SCs) and can reproduce the structure and phenotype of all epithelial cells of the original tissue in a 3D system [13,14]. Since the time of the first publications, PDOs have been established for many human tissues including prostate [6,9,15]. Although PDOs can originate from a single cell and, unlike PDXs, do not have the vascular and stromal microenvironment, they reproduce all the traits of the original tissue, including its genetic and phenotypic characteristics [16]. One of the important advantages of PDOs in translational research is their ability to rapidly expand so that they can be used in high-throughput drug screening [17]. Molecular profiling of PDOs can be instrumental in the identification of disease prognostic markers and the prediction of tumor response to personalized drug treatment [18,19,20].

Cells from normal and cancerous prostate tissue can grow with a low efficiency in 2D cultures. Primary prostate cell cultures (PCCs) form as a result of the interaction between cells with diverse phenotypes. In contrast to primary cell cultures from other organs (bone marrow, adipose tissue, endometrium, bladder, and others) characterized by the fast appearance and dominance of mesenchymal cells [21,22,23,24], human primary PCCs have a heterogeneous cell composition. Thus, cultured explants of human prostate tissue were found to contain the cells with the epithelial morphology, while the cells with the stromal features appeared after 21 days of culturing [25]. Human 2D prostate cell cultures derived from the tissue treated with collagenase originally consisted of the cells expressing epithelial markers, but then became uniformly fibroblast-like at passage 5. The cells in the culture expressed vimentin, with a small number of cells co-expressing alpha smooth muscle actin (α-Sma) and vimentin (myofibroblasts) or α-Sma and desmin (myoblasts) [26]. Prior to expansion, the primary culture of prostate stromal cells contained a small population of mesenchymal stem cells (MSCs) that expressed mesenchymal markers CD90, CD73, and CD105 [27,28]. Such an unusual non-mesenchymal composition of human primary PCCs in vitro corresponds to the tissue structure of the human prostate, which is composed mostly of muscle cells and myofibroblasts [29].

We suggested that the primary 2D cell cultures obtained from normal or cancerous prostate tissues might contain a small amount of tissue-specific normal or cancerous stem cells (SCs), whose descendants express epithelial and tumor markers. To verify this hypothesis, we used human 2D PCCs and PDOs derived from the cancerous and surrounding normal prostate tissues from patients with localized PCa. The cells were cultured for six passages, so the PCCs were designated as early (passages one to three) and late (passages four to six). The cells from the cancerous and surrounding normal tissues and derived PCCs were used to prepare PDOs. The cells from the prostate tissue, PCCs, and PDOs were assessed for the epithelial basal (CK5, p63) and luminal (CK18, AR1) markers, mesenchymal (vimentin) marker, and cancer (AMACR, TMPRSS2-ERG, and histone methyltransferase EZH2) markers by RT-PCR. The PCR products obtained by amplification of the prostate cancer marker TMPRSS2-ERG on cDNA from the prostate cancer tissue (PCaT) and derived early PCCs and PDOs were cloned and sequenced. Literature data show that the level of EZH2 is stably elevated in aggressive PCa but not in localized PCa [30]. AMACR is commonly used as a marker of all PCa stages [31]. TMPRSS2-ERG is a specific PCa marker, although its expression has been found in less than 50% of patients with localized PCa [32]. Both TMPRSS2-ERG and EZH2 are included in the Catalogue of Somatic Mutations in Cancer [33], however the whole-genome sequencing of PCa did not designate these genes as cancer drivers [34]. On the other hand, it was established that the EZH2 expression is regulated by the pRb-E2F1 signaling and the loss of pRb leads to an increase in the EZH2 level [35]. AMACR regulates fatty acids β-oxidation and energy metabolism, which is activated in PCa and other malignancies. However, the mechanisms of AMACR hyperproduction in cancer are still unknown [31]. Evaluation of the expression of these cancer markers may facilitate the use of PCCs as a preclinical model of PCa.

We found that the cells of early PCCs and corresponding PDOs expressed epithelial and cancer markers (AMACR, TMPRSS2-ERG, and EZH2). The upregulation of the EZH2 expression in the cancerous tissue from the localized PCa tissue derived early PCCs and PDOs was detected at the same frequency as for other tumor marker AMACR. These results suggest that EZH2 can be considered as a marker of localized PCa, while early PCCs can be used as an appropriate preclinical model for studying the pathogenesis of PCa.

## 2. Results

### 2.1. Experimental Design

Each of the collected samples of cancerous and normal prostate tissue was divided into three equal parts. One part was enzymatically treated to prepare PDOs and early PCCs. The second part was used to extract proteins for electrophoresis and immunoblotting. The third part was used to extract total RNA for RT-PCR. Cells of the early PCCs served to prepare late PCCs and both early and late PCCs were used for the preparation of PDOs and total RNA extraction. The cells of early and late PCCs were analyzed under a microscope in transmitted light and by immunofluorescence staining. Some RT-PCR products amplified from the prostate tissue, early PCCs, and derived PDOs were cloned and sequenced (Figure 1).

### 2.2. Immunoblotting Analysis of Prostate Tissue for Cancer Markers and Immunofluorescence Staining of Early and Late PCCs for Epithelial and Cancer Markers

First, we evaluated cancerous and normal prostate tissue samples for the presence of cancer markers AMACR and EZH2. PCaT samples were positive for these markers (similar to the T98G glioblastoma cells that served as a positive control [31]), while normal prostate tissue did not express them (Figure 2a).

### 2.3. Morphological Characterization of Cells in the Early and Late PCCs and in PDOs Prepared from Prostate Tissue and PCCs

Early PCCs cultured in the SCBM Stromal Cell Medium (Lonza, Basel, Switzerland) originally consisted predominantly of polygonal clusters of cells tightly adhering to each other, and to the dish bottom, but also contained a small number of fibroblast-like cells growing separately (Figure 3i(a)). Over the course of culturing, the number and the size of the polygonal cell clusters decreased (Figure 3i(b)). The clusters completely disappeared in late PCCs (passage 4–6). In contrast, the number of fibroblast-like cells progressively increased; in the late PCCs, the fibroblast-like cells dominated and eventually became the only cell type (Figure 3i(c)). On day 10, the PDOs derived from the prostate tissue were round-shaped homogenous structures (Figure 3i(d)). The organoids prepared from early PCCs were more heterogeneous and smaller in size in comparison with the tissue-derived PDOs and contained a few fibroblast-like structures with one or two outgrowths (Figure 3i(e)). Some PDOs prepared from late PCCs contained round-shaped organoids that were smaller than those prepared from the prostate tissue and early PCCs; however, they typically included a significant amount of fibroblast-like organoids with a few outgrowths (Figure 3i(f)). To evaluate the difference in the number and size of the PDOs prepared from different sources, we counted the number of round-shaped small, medium, and large organoids and of the atypical fibroblast-like organoids, using the ImageJ program (Figure 3ii). Quantitative analysis showed that the total number of organoids that have originated from the prostate tissue and early and late PCCs was similar; however, the PDOs prepared from PCCs were more heterogenous than the tissue-derived ones. They progressively decreased in size and formed the fibroblast-like structures with a few outgrowths (Figure 3ii(f)). The total amount of such organoids differed from sample to sample; on average, they represented 50% of the total population of PDOs derived from the late PCCs (Figure 3ii).

### 2.4. RT-PCR Evaluation of Epithelial and Cancer Markers in Prostate Tissue, Early and Late PCCs, and Derived PDOs

To characterize the phenotypes of the epithelium in PCaT and normal prostate tissue and in derived early and late PCCs and PDOs, we assessed the expression of basal (CK5, p63) and luminal (CK18, AR1) epithelial markers and prostate cancer markers (AMACR and EZH2) by RT-PCR. All analyzed cancerous and normal prostate tissue samples obtained from eighteen patients, as well as derived PDOs, expressed the basal and luminal epithelium markers (Figure 4a). PCaT samples and derived PDOs expressed cancer markers AMACR, EZH2 and TMPRSS2-ERG (Figure 4a–c and Figure 5b). In the early PCCs and PDOs derived from them, the expression of all studied markers was very similar to that in the tissues (Figure 4a,b). No epithelial markers were detected in late PCCs prepared from the cancerous and normal tissues and in the derived PDOs; however, all analyzed cells expressed vimentin (Figure 4c). Additionally, late PCCs and PDOs derived from PCaT expressed AR1, AMACR, and EZH2 in contrast to the cell cultures prepared from normal prostate tissue (Figure 4c).

Detection of epithelial cancer markers in the homogenously stromal cell population was unexpected. To confirm this observation, we assessed the expression of EZH2 and AMACR by late PCCs using immunofluorescence staining and found that late PCCs derived from PCaT expressed these markers, while cell cultures derived from the normal prostate tissue did not. Interestingly, about 50% of PCaT samples and derived organoids expressed another cancer marker, TMPRSS2-ERG, in addition to AMACR and EZH2. The expression of this marker was also detected in early PCCs and PDOs derived from them (Figure 5b).

### 2.5. Cloning and Sequencing of RT-PCR Products of the Prostate Cancer Marker TMPRSS2-ERG

The fusion of the 5′ regulatory region of the androgen-responsive gene TMPRSS2 with the downstream ERG gene has been found in most PCa cases [32]. Based on this observation and the fact that the TMPRSS2-ERG fusion products are more common in moderately to poorly differentiated PCa vs. well-differentiated PCa, we examined the expression of the TMPRSS2-ERG fusion mRNAs in PCaT, PCCs, and PDOs. Gel electrophoresis of the RT-PCR products showed that the amplification of cDNA from PCaT, PCCs, and PDOs usually yielded a single band, although for some PCCs and PDOs, amplification produced several different products, as was later confirmed by their cloning and sequencing (Figure 5b(i,ii)).

Although RT-PCR products for PCCs were seen as three bands in the agarose gel, cloning and sequencing of the amplified fragments revealed the presence of four TMPRSS2-ERG and one TMPRSS2-PPP6R1 fusion transcripts in PCCs variant 2, three transcripts in PDOs (180- and 56-b TMPRSS2-ERG transcripts and one TMPRSS2-PPP6R1 transcript) and two TMPRSS2-ERG transcripts (180 and 56 bp) in PCCs variant 1, indicating the presence of low-abundant transcripts among the reaction products (Figure 5b,c,e). These results demonstrated the advantage of the cloning/sequencing approach for evaluating the fused transcripts. Furthermore, two transcripts contained the G > A substitution in the CGG triplet, thus turning it into CAG (Figure 5d). Analysis of the splicing site in all known TMPRSS2-ERG variants demonstrated that the CAG is the most common last TMPRSS2 triplet in the fusion gene transcripts, suggesting its involvement in alternative splicing. We found two new fusion mRNA transcripts among the cloned fragments originating from PCCs and PDOs. One consisted of the truncated exon 1 of TMPRSS2 (1–33 bp) fused with the ERG exon 4 at the canonical position, and the other consisted of the truncated TMPRSS2 exon 1 (1–21 bp) fused to the ERG exon 4 at position 388. Apparently, active mutagenesis occurring in PCCs and PDOs might lead to erroneous splicing. This is the first observation on the contribution of mutations to the diversity of TMPRSS2-ERG fusion variants. Meanwhile, fusion products amplified on the PCaT mRNA were represented by a single band irrespective of the method used for their identification. Sequence analysis showed that this was the most common TMPRSS2-ERG fusion variant found in the PCa samples. In addition to the novel chimeric TMPRSS2-ERG transcripts produced by the intrachromosomal translocation/deletion events, we detected a recombinant TMPRSS2-PPP6R1 transcript resulting from the interchromosomal translocation (Figure 5e). This transcript, which appeared to be a low-abundant fusion variant, was identified by cloning/sequencing in both PCCs and PDOs.

### 2.6. Expression of AMACR, EZH2 and TMPRSS2-ERG in Cancerous and Normal Prostate Tissues and Early PCCs

Comparing the results on the expression of AMACR, EZH2, and TMPRSS2-ERG cancer markers obtained by immunoblotting, immunofluorescence staining, and RT-PCR showed that AMACR and EZH2 were expressed in all localized cancer samples (100%) in contrast to TMPRSS2-ERG, which was expressed only in some samples (Table 1). This indicates that EZH2 cannot be used as a marker of transition from the dormant to aggressive cancer. On the other hand, all three markers were expressed in PCaT and derived early PCCs, suggesting that early PCCs can be used as a preclinical model to study the pathogenesis of PCa.

## 3. Discussion

The development of PCa as one of the most common types of cancer and its treatment have become the focus of oncological research. PCa development includes two stages—indolent cancer and aggressive metastatic castration-resistant PCa (CRPC). Although the two PCa stages are mechanistically different, they are closely related to each other. PCa can persist in the dormant form until the natural end of life or can transform to aggressive CRPC already at the initial stage of its development [36,37]. The early markers for such transformation are yet unknown, but some publications point out that clinically localized PCa with upregulated EZH2 expression shows a poorer prognosis [30]. EZH2 regulates self-renewal of different types of SCs through H3K27 trimethylation that represses the differentiation of normal and cancer SCs [38,39]. The content of EZH2 was found to be elevated in breast cancer SCs in comparison with normal tissue-specific SCs, suggesting that EZH2 may initiate cancer growth by blocking the differentiation of somatic SCs [38,40].

CRPC is now a lethal disease, as the mechanism and the markers of dormant PCa transformation into aggressive metastatic cancer still remain unknown. Investigating this transition requires the establishment of affordable preclinical models that would recapitulate the features of PCa development and can be used by many researchers. We selected primary prostate cell cultures (PCCs) as a model of PCa development. Our preliminary experiments showed that in contrast to primary cell cultures from other tissues, PCCs are formed by epithelial cells [41]. Here, we confirmed that primary PCCs consist of epithelial cells (Figure 2ii), which is promising because a single prostate epithelial cell can form a 3D organoid culture in Matrigel [15]. It is generally accepted that PDOs derived from cancerous and normal prostate tissues contain SCs that retain for a long period of time the ability to differentiate between basal and luminal prostate epithelium and, therefore, can be used for PCa studies [6,42]. Our suggestion was verified by comparing PDOs prepared from the prostate tissue and early and late PCCs. All these cell sources contained prostate SCs and were able to form PDOs (Figure 3i). However, quantitative evaluation of different-size PDOs derived from these cell sources showed that the number of large-size organoids continuously decreased from prostate tissue to late PCCs. We also observed the formation of an increasing number of atypical fibroblast-like organoids in PCCs (Figure 3i,ii), indicating that the population of prostate specific SCs that formed large-size PDOs continuously decreased and lost its abilities.

RT-PCR revealed that the expression profiles of the basal epithelial, luminal epithelial, mesenchymal, and PCa markers in the cancerous and normal early PCCs were very similar in comparison to those in the tissues these PCCs were derived from (Figure 4). In contrast, late PCCs lost expression of epithelial markers, although PCCs prepared from PCaT continued to express androgen receptor (AR1) and prostate cancer markers AMACR and EZH2 (Figure 4c). This was surprising because late PCCs represent homogenous populations of mesenchymal cells, while AR1, AMACR, and EZH2, as we believe, are the markers of epithelial cells. To verify these results, we evaluated the expression of AMACR and EZH2 with late PCCs derived from the cancerous and normal prostate tissues using immunofluorescent staining. The results of immunofluorescence staining supported the RT-PCR data: late PCCs produced from PCaT expressed AMACR and EZH2 in contrast to the cell cultures obtained from the normal tissue (Figure 4d).

We also observed diverse TMPRSS2-ERG fusion transcripts in PCCs and PDOs derived from PCaT, including new chimeric variants resulting from the intra- and interchromosomal translocations. Expression of a TMPRSS2-ERG fusion mRNA variant in which TMPRSS2 ATG was in frame with the ERG exon 4 was shown to be associated with aggressive PCa [43]. Interestingly, this fusion transcript was only found in PCCs variant 1 but not in the other samples, including PCaT (Figure 5b), which may be due to the presence of several different cancer SCs clones in the same PCa sample. Finding two new TMPRSS2-ERG fusion transcripts associated with the G > A substitution in the CGG triplet indicated the importance of the CAG site for the TMPRSS2-ERG splicing and its possible contribution to the diversity of fusion variants (Figure 5d). In addition to novel chimeric TMPRSS2-ERG transcripts resulting from the intrachromosomal translocation/deletion events, we detected a recombinant TMPRSS2-PPP6R1 chimera produced by the interchromosomal translocation (Figure 5e). This transcript, which appeared to be a low-abundant fusion variant, was identified by cloning/sequencing in both PCCs variants 1 and 2. Therefore, application of the cloning/sequencing approach might expand the study of existing rare variants of fusion transcripts. It has been shown that soluble guanylyl cyclase (sGC) is strongly associated with the TMPRSS2-ERG fusion in clinical PCa cohorts and that the sGC-mediated NO-cGMP pathway is a critical downstream effector of ERG in promoting cancer cell proliferation and tumorigenesis [44]. Hence, targeting NO-cGMP signaling could be a new therapeutic strategy for the treatment of PCa related to the TMPRSS2-ERG gene fusion and might be of interest in preclinical and clinical PCa studies. The association of newly discovered low-abundant TMPRSS2-ERG chimeras with PCa aggressiveness remains to be elucidated.

Prostate tumor is a malignant tissue surrounded and infiltrated with stromal cells commonly designated as cancer-associated fibroblasts (CAFs). CAFs secrete cytokines and growth factors and strongly promote PCa growth compared to normal prostate fibroblasts [45]. In contrast to PCa epithelial cells, the expression of the androgen receptor in the prostate stroma is downregulated and is inversely related to a poor PCa outcome [46]. The deletion of stromal AR in smooth muscle cells promoted tumor progression in mice [47]. Presumably, there is a reciprocal relationship between CAFs, AR signaling, and PCa progression. We suggest that the elimination of epithelial cancer cells in late PCCs may enhance the AR signaling in surrounding CAFs, while activation of AR signaling can induce the expression of EZH2 and AMACR [48].

## 4. Material and Methods

### 4.1. Prostate Tissue Samples

Prostate tissue samples were obtained from 23 patients who had undergone radical prostatectomy for prostate cancer (PCa) at the Urology Department of the Second City Clinical Hospital, St. Petersburg according to the protocols approved by the Hospital Ethical Committee. Prostate tissue samples (~100 mg) containing a tumor node or normal tissue from the same prostate layer were placed in tubes with 5 mL of sterile saline containing 50 μg/mL gentamicin (Biolot, St. Petersburg, Russia). Each sample was cut into three parts with sharp scissors. One part was used for protein extraction followed by protein electrophoresis and immunoblotting. The second part was used for total RNA extraction, cDNA synthesis, and RT-PCR. The third part was treated with collagenase to prepare a cell suspension that was used to produce PCCs and PDOs.

### 4.2. Establishment of Prostate Cell Cultures

Cancerous and normal prostate tissue samples were placed in 100 mm cell culture dishes in 10 mL of sterile phosphate buffered saline (PBS), cut with sterile scissors into 1 mm pieces, transferred into 15 mL conical plastic tubes (TPP, Novosibirsk, Russia), and subjected to the enzymatic treatment in 1 mL of a solution containing 5 mg of type II collagenase (Life Technologies, Carlsbad, CA, USA) in 1 mL of adDMEM/F12 containing 50 μg/mL of penicillin and streptomycin, 10 mM of HEPES, and GlutaMAX (adDMEM/F12+++ medium; Life Technologies, Carlsbad, CA, USA), supplemented with 10 μM Y-27632 Rho kinase inhibitor (Abmole Bioscience, Houston, TX, USA) and 1 nM dihydrotestosterone (Sigma, Saint Louis, MO, USA) [49]. The tubes were incubated in a PST-60 HL-4 rotary shaker (Biosan, Riga, Latvia) at 230 rpm for 18 h at 37 °C. Next, the cell suspension was centrifuged at 250× *g* for 5 min. The pellet was washed in 5 mL of adDMEM/F12 medium, filtered through a nylon sieve with 70-μm pores (Corning, Tewksbury, MA, USA), and precipitated at 250× *g* for 5 min. The pellet was resuspended in 1 mL of TrypLE Express solution (Life Technologies, Carlsbad, CA, USA) and incubated for 1 h in a PST-60 HL-4 rotary shaker (Biosan, Riga, Latvia) at 230 rpm at 37 °C. Five milliliters of adDMEM/F12 was added to the suspension. The cells were pelleted by centrifugation at 250× *g* for 5 min, and the pellet was resuspended in 500 μL of adDMEM/F12+++. To count the cells, 5 μL of the cell suspension was mixed with 45 μL of 1% acetic acid and 50 μL of 2% trypan blue (Biolot, St.Petersburg, Russia). The cells were counted in a Goryaev’s chamber. To obtain primary stromal cultures, 5 × 10^5^ cells from the prostate cell suspension were added to 4 mL of SCBM (Lonza, Basel, Switzerland) and plated into 60 mm culture dishes (TPP, Novosibirsk, Russia). Subsequent passages were performed at 90% cell density by splitting the cells on the same plates at a ratio of 1:2. The cells were cultured for 6 passages to produce early (passages 1 to 3) and late (passages 4 to 6) PCCs. The images of cell cultures were obtained using an Axiovert 200 M microscope (Carl Zeiss, Oberkochen, Germany) equipped with a DFC420 camera in transmitted light or in the phase contrast mode using a 20×/0.5 objective; the image size was 1728 × 1296 pixels.

### 4.3. Production, Culturing, and Subculturing of PDOs Derived from Cancerous and Normal Prostate Tissues

Prostate tissue cells (1 × 10^5^) obtained by tissue treatment with collagenase were resuspended on ice in 80 μL of adDMEM/F12+++ mixed with 80 μL of Matrigel (BD, San Jose, CA, USA) preliminarily melted on ice. The Matrigel/cell mixture (40 μL) was placed into each of four wells of a 24-well culture plate (Corning, Tewksbury, MA, USA); the plate was turned over and placed in a CO_2_ incubator for 15 min. Next, 500 μL of the PDO culturing medium of the following composition was added to each well: 1.0 mL of B27 (Life Technologies, Carlsbad, CA, USA), 500 μL of 1 M nicotinamide in PBS (Sigma, Saint Louis, MO, USA), 125.0 μL of 500 mM N-acetylcysteine (Sigma, Saint Louis, MO, USA), 0.5 μL of EGF solution (0.5 μg/mL in PBS + 0.1% BSA) (PeproTech, Cranbury, NJ, USA), 5 μL of 5 mM A83-01 in DMSO (Tocris Bioscience, Ellisville, MO, USA), 50 μL of Noggin (100 μg/mL in PBS + 0.1% BSA) (PeproTech, Cranbury, NJ, USA), 50 μL of R-spondin 1 solution (500 μg/mL in PBS + 0.1% BSA) (R&D Systems, Minneapolis, MN, USA), 50 μL of 1 μM dihydrotestosterone in ethanol (Sigma, Saint Louis, MO, USA), 5 μL of FGF2 solution (50 μg/mL in PBS + 0.1% BSA) (PeproTech, Cranbury, NJ, USA), 5 μL of FGF10 solution (0.1 μg/mL in PBS + 0.1% BSA) (PeproTech, Cranbury, NJ, USA), 5 μL of 10 mM of E2 prostaglandin in DMSO (Tocris Bioscience, Ellisville, MO, USA), 16.7 μL of 30 mM SB202190 in DMSO (Sigma, Saint Louis, MO, USA), and up to 50 mL of adDMEM/F12+++ (Life Technologies, Carlsbad, CA, USA).

At the beginning of the PDO culturing, 5 μL of 100 mM Y-27632 reagent (Abmole Bioscience, Houston, TX, USA) was added to the culture medium [49]. After 10 days, organoids from each well of a 24-well plate were suspended in their own culture medium, transferred into 15-mL plastic tubes, and precipitated by centrifugation for 5 min at 250× *g*. The pellets were resuspended in 1 mL TrypLE with 10 μM Y-27632 and the tubes were placed in an orbital shaker for 5 min at 37 °C. TrypLE was inactivated by adding adDMEM/F12+++ containing 5% FBS and the cells were centrifuged at 200× *g* for 5 min. The supernatant was removed, and the cell pellet was resuspended in 80 μL of adDMEM/F12+++ and mixed with 80 μL of Matrigel. The Matrigel/cell mixture (40 μL) was placed in the center of a well of a 24-well plate. The plate was turned over and placed in a CO_2_ incubator for 15 min; next, 500 μL of PDO culturing medium containing 10 μM Y-27632 Rho kinase inhibitor was added to each well. The medium was exchanged every 2–3 days [49].

### 4.4. Establishment and Quantification of PDOs from Early and Late Prostate Stromal Cell Cultures

The cells of early and late PCCs grown in 60 mm plastic dishes were washed twice with 1 mL of PBS, separated from the plastic surface by treating with 0.5 mL of 0.25% Trypsin/Versene solution at 37 °C for 5 min, resuspended in 5 mL of PBS, and pelleted by centrifugation at 250× *g* for 5 min. The pellet was resuspended in 500 μL of adDMEM/F12+++ and the cells were counted in a Goryaev’s chamber. The cells (2 × 10^5^) were resuspended on ice in 20 μL of adDMEM/F12+++ and mixed with 140 μL of Matrigel (BD, San Jose, CA, USA) preliminarily melted on ice. The Matrigel/cell mixture (40 μL) was placed into each of the four wells of a 24-well culture plate (Corning, Tewksbury, MA, USA) and the plate was treated as described above.

To estimate difference in the number and size of PDOs prepared from the prostate tissue and early and late PCCs, we separately evaluated three groups of organoids differing in their diameter: small (10–25 μM), medium (26–50 μM), and large (51–100 μM). We counted the total number of organoids, the number of organoids in each group, and the number of atypical fibroblast-like organoids, which formed mostly in the PDOs prepared from late PCCs. For quantification, we obtained 15 images in the transmitted light of PDOs over 10 days of culturing using an Axiovert 200M microscope (Carl Zeiss, Oberkochen, Germany) and a 20×/0.5 objective; the image size was 1728 × 1296 pixels. The images were quantified with the ImageJ program; the data for each PDOs type were obtained from three different samples and used to calculate the arithmetic mean and standard deviation using Microsoft Office Excel, 2010.

### 4.5. Reverse Transcription Polymerase Chain Reaction (RT-PCR)

The PCR mixture contained 1 μL of 10 mM dNTP mixture (Evrogen, Moscow, Russia), 2.5 μL of 10× *Taq* DNA polymerase buffer, 2.5 μL of 10 mM MgCl2, 1 μL of forward and reverse primers each (Table 2), 2 μL of DNA, 0.25 μL of *Taq* polymerase (Alkor Bio, St.Petersburg, Russia), and water up to 25 μL. Amplification was carried out in a Bio-Rad T100 thermal cycler (Hercules, CA, USA) using the following temperature regime: initiating denaturation melting, 1 min, 94 °C; denaturation, 15 s, 94 °C; annealing, 30 s, 58 °C; elongation, 30 s, 72 °C (35 cycles); final elongation, 10 min, 72 °C. PCR amplification products were subjected to electrophoresis in 1.2% agarose gel with ethidium bromide, and the size of individual amplicons was determined by comparing their mobility with 100-bp DNA ladder (Invitrogen, Carlsbad, CA, USA) in images obtained with a BioRad ChemiDoc XRS+ (Hercules, CA, USA) (Table 1). PCR-products containing the TMPRSS2-ERG mutant fusion gene were cloned and sequenced.

### 4.6. Sodium Dodecyl Sulfate (SDS)-Polyacrylamide Gel Electrophoresis (PAGE)

To prepare protein extracts, asynchronously growing cells were washed twice with PBS, separated from plastic with a scraper, sedimented by centrifugation (4000× *g*, 1 min), and incubated for 30 min on ice in three volumes of lysing solution containing 50 mM Tris-HCl (pH 8.0), 150 mM NaCl, 1% NP-40, 0.1% SDS, 0.1 mM PMSF, and protease inhibitor cocktail (dilution, 1:100; Sigma, Saint Louis, MO, USA). The cell extracts were centrifuged for 15 min at 14,000× *g* and 4 °C, and the supernatants were frozen and stored at −80 °C until used in the experiments. Protein concentration was determined by the Bradford method. Before loading on the gel, the protein amount in the samples was equalized by adding an appropriate amount of 4× loading buffer (4% SDS, 20% glycerol, 200 mM dithiothreitol, 120 mM Tris-HCl (pH 6.8), 0.002% bromophenol blue); the samples were incubated at 95 °C for 5 min and loaded on the gel (50 μg of total protein per lane). Electrophoresis was performed in 10% SDS-PAAG at 30 mA per gel.

### 4.7. Immunoblotting

Electrophoretically separated proteins were transferred to a PVDF membrane (Thermo Fisher Scientific, Waltham, MA, USA) by semi-dry electrotransfer in a Hoefer Semaphore TE 77 chamber (Hoefer, Holliston, MA, USA) at 90 mA for 1 h. Before the transfer, the membrane was placed in methanol for 1 min and then incubated in the transfer buffer (48 mM Tris-Cl, 39 mM glycine, 10% methanol, 0.03% SDS). After the transfer, the membrane was blocked with 5% fat-free milk in TBST buffer (150 mM NaCl, 10 mM Tris-HCl (pH 8.0), 0.05% Tween 20) for 1 h to prevent nonspecific binding. Next, the membrane was washed in TBST (five times for 5 min) and treated with the primary antibodies overnight at 4 °C, followed by the incubation with the species-specific horseradish peroxidase (HRP)-conjugated secondary antibodies for 2 h at room temperature. The antibodies were diluted in a blocking solution as recommended by the manufacturers. Proteins on the membrane were visualized using a Clarity Western ECL Substrate reagent kit (Bio-Rad, Hercules, CA, USA) and documented with a ChemiDoc Touch Imaging System (Bio-Rad, Hercules, CA, USA).

### 4.8. Immunofluorescent Staining

Cells grown on round glass coverslips (diameter, 9 mm; Biovitrum, St.Peterburg, Russia) were transferred into 35 mm dishes, washed once with PBS for 5 min, fixed with 4% paraformaldehyde for 15 min and then with 70% ethyl alcohol overnight at 4 °C, treated with 0.2% Triton X-100 for 10 min, and washed with PBS twice for 5 min. Nonspecific antibody binding was blocked for 1 h with a solution containing 3% BSA and 0.1% Tween 20. Next, specific antibodies (dilution, 1:50–200) were applied to the cells in a blocking solution for 1 h at room temperature. The cells were then washed three times for 5 min with PBS, and secondary anti-rabbit IgG conjugated with Cy5 or anti-mouse IgG conjugated with Cy3 were applied for 1 h at room temperature. The coverslips with the cells were washed three times for 5 min with PBS and embedded in Anti-Fade medium (BioRad, Hercules, CA, USA) to reduce nonspecific fluorescence; the medium contained DAPI for DNA staining. Immunofluorescence images were obtained with a Leica scanning microscope (Leica Microsystems, Wetzlar, Germany) using 405, 543, and 633-nm lasers and 40× or 63× objectives.

### 4.9. TA Cloning and DNA Sequencing

TMPRSS2-ERG translocation variants were amplified with *Taq* DNA polymerase, purified with a QIAquick PCR Purification Kit (Qiagen, Düsseldorf, Germany), and cloned into the pCR2.1-TOPO vector using a TOPO TA Invitrogen Cloning kit. The resulting plasmids were used for the transformation of *Escherichia coli* XL1-blue cells; positive clones were grown in LB medium overnight. The recombinant plasmids were isolated by the DM miniprep method [50] and sequenced in both directions using M13 forward and reverse primers (Invitrogen) and ABI BigDye Terminator Cycle Sequencing Kit v3.1 (Thermo Fisher, Waltham, MA, USA) with a Gene Amp 9700 PCR System (Thermo Fisher, Carlsbad, CA, USA). The sequences were detected with an ABI 3130XL Genetic Analyzer (Applied Biosystems, Foster City, CA USA). The sequences of TMPRSS2 (NM_005656) and ERG (NM_004449) were analyzed with the BLAST software (National Center for Biotechnology Information, NIH, GOV, USA).

### 4.10. Antibodies

The following antibodies were used: anti-EpCAM (Abcam, Cambridge, UK), anti-PanCK (Thermo Scientific, Waltham, MA, USA), anti-α-Sma, (Thermo Fisher Scientific, Waltham, MA, USA), anti-vimentin (Sigma, Saint Louis, MO, USA), mouse anti Ezh2 and mouse anti Gapdh antibodies (Santa Cruz Biotech, Dallas, TX, USA); mouse anti-human AMACR antibody was produced in our laboratory [31]. Secondary HRP-conjugated anti-rabbit and anti-mouse IgG were purchased from Cell Signaling (Danvers, MA, USA) and BioRad (Hercules, CA, USA), respectively. Cy3-labeled goat anti-mouse IgG (H+L) and Cy5-labeled goat anti-rabbit IgG (H+L) used as the secondary antibodies in immunofluorescence staining were from Invitrogen, (Carlsbad, CA, USA).

### 4.11. Statistical Analysis

Western blot and immunofluorescence experiments were conducted in triplicate. ImageJ software was used to quantify the light microscopy data. The data were processed and presented as mean ± SD using Excel 2010. The groups were compared using the Student’s *t*-test (*p* < 0.05, *p* < 0.01).

## 5. Conclusions

Primary prostate cell cultures (PCCs) were formed by epithelial cells that were progressively replaced by mesenchymal cells, resulting in the formation of homogeneous stromal cell populations at passage six. Early PCCs (passages 1–3) from cancerous and normal tissues contained tissue-specific cancer and normal stem cells, respectively, that formed patients derived organoids (PDOs) in the Matrigel-based 3D cultures. Similar to prostate cancer cells and derived PDOs, early PCCs and PDOs derived from them expressed cancer markers AMACR, TMPRSS2-ERG, and EZH2. Our results suggest that early 2D PCCs can be used as a preclinical model to study the pathogenesis of localized PCa.

## Figures and Tables

**Figure 1 ijms-24-02830-f001:**
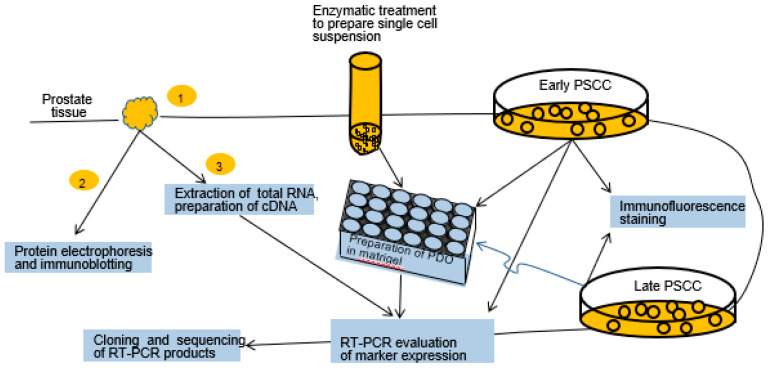
General experimental design. The samples of human prostate, cancerous and surrounding normal tissues, from PCa patients treated with radical prostatectomy were divided into three equal parts. One part was analyzed by Western blotting; the second part was used for the total RNA extraction and cDNA preparation for the analysis of cell markers by RT-PCR. The third part was enzymatically treated for the production of early and late cell cultures (PCCs) and PDOs. The cells of early and late PCCs were used for the microscopic analysis and immunofluorescence staining. The products of amplification of the TMPRSS2-ERG marker on prostate tissue, early PCCs, and derived PDOs were cloned and sequenced.

**Figure 2 ijms-24-02830-f002:**
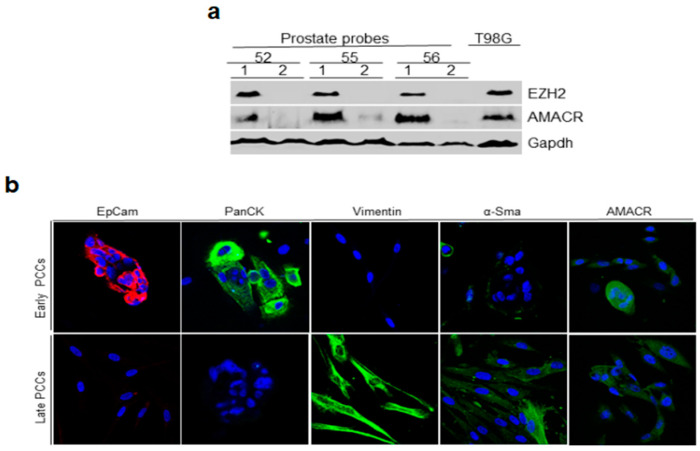
Immunoblotting and immunofluorescence staining of the epithelial, mesenchymal, and cancer markers in the prostate tissue and early and late PCCs derived from PCaT. (**a**) Immunoblotting of prostate cancer markers AMACR and EZH2 in cancerous (1) and normal (2) tissues. Protein extracts (30 µg of total protein per lane) were loaded onto 10% polyacrylamide gel. The extract of T98G cells served as a positive control, Gapdh was used as a loading control; (**b**) early and late PCCs were stained with antibodies against epithelial (EpCam, PanCK), mesenchymal (vimentin, α-Sma), and cancer (AMACR) markers. Images were captured with a digital scanning confocal microscope (Leica Microsystems, Wetzlar, Germany) using 405, 488, 543 and 633-nm lasers, objective, ×63.

**Figure 3 ijms-24-02830-f003:**
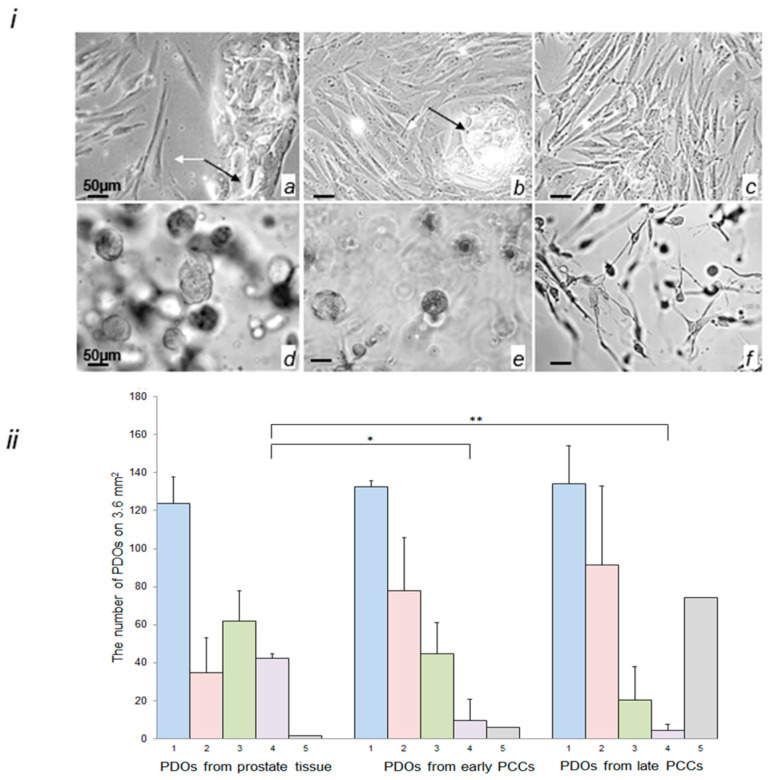
Morphological characteristics of early and late PCCs and PDOs derived from prostate tissue and early and late PCCs. (**i**) Representative images of early PCCs (**a**,**b**), late PCCs (**c**), PDOs derived from prostate tissue (**d**), and PDOs derived from early (**e**) and late (**f**) PCCs. Images were obtained using an Axiovert 200M microscope (Carl Zeiss, Oberkochen, Germany) equipped with a DFC420 camera in the phase contrast mode; epithelial cells—black arrows; mesenchymal cells—white arrows; objective, 20×/0.5; image size, 1728 × 1296 pixel. (**ii**) The number of different-size PDOs derived from the prostate tissue, early PCCs, and late PCC (n = 3; 20 images per group; mean ± SD); 1—total number of PDOs, 2—the number of small-size PDOs (10–25 μM), 3—the number of medium-size PDOs (26–50 μM), 4—the number of large-size PDOs (51–100 μM), 5—the number of fibroblast-like PDOs, not included in the total number of PDOs; * *p* < 0.05, ** *p* < 0.01.

**Figure 4 ijms-24-02830-f004:**
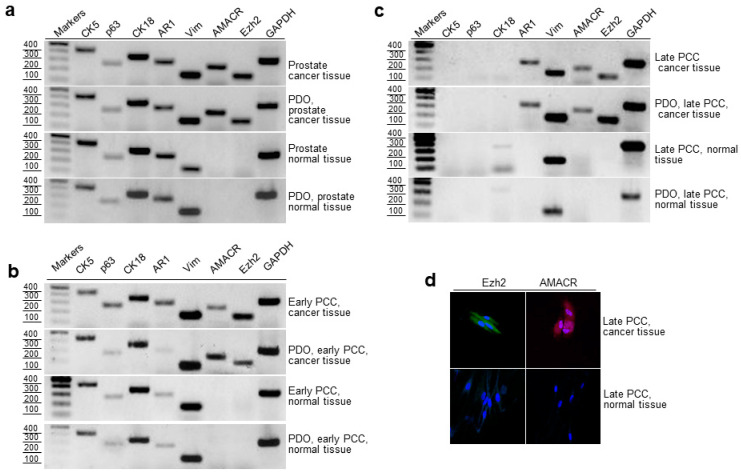
RT-PCR evaluation of epithelial, mesenchymal, and cancer markers in the cancerous and normal prostate tissue, PCCs, and PDOs. (**a**) Expression of epithelial (CK5, CK18, p63, AR1), mesenchymal (vimentin), and cancer (AMACR and EZH2) markers in (**a**) PCaT and surrounding normal tissue and derived PDOs; (**b**) early PCCs and derived PDOs; (**c**) late PCCs and derived PDOs; (**d**) Immunofluorescence staining of EZH2 and AMACR in late PCCs derived from the cancerous and normal prostate tissues (nuclei were stained with DAPI). The images were captured with a scanning confocal microscope (Leica Microsystems, Wetzlar, Germany) using 405, 488, and 633-nm lasers, objective, ×40.

**Figure 5 ijms-24-02830-f005:**
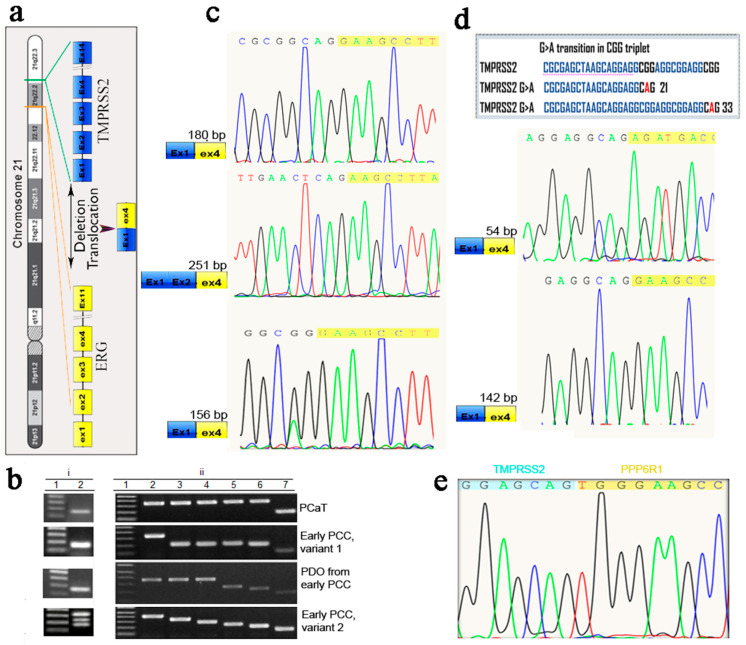
Verification of expression of prostate cancer marker TMPRSS2-ERG in PCaT, early PCCs and derived PDOs by RT-PCR followed by cloning and sequencing of the amplification products. (**a**) Diagram of TMPRSS2-ERG gene fusion on chromosome 21. (**b**) Detection of TMPRSS2-ERG fusion transcripts by (**i**) RT-PCR and (**ii**) T/A cloning of the amplified RT-PCR products. (**i**) Amplification with a forward primer located in the TMPRSS2 exon 1 and reverse primer located in the ERG exon 4. RT-PCR products were resolved by agarose gel electrophoreses; lane 1—100-bp DNA ladder, lane 2—RT-PCR amplification products in the samples designated on the right. Early PCCs, variants 1 and 2, were derived from different parts of a tissue sample from the same patient. (**ii**) T/A cloning of the products of amplification of the TMPRSS2-ERG transcripts and analysis of bacterial clones; lane 1—1 kb plus DNA ladder, lanes 2–6—amplification products corresponding to the TMPRSS2-ERG transcripts amplified from the samples indicated on the right, lane 7 contains the DNA fragment amplified on the control vector. The PCR products were amplified using M13 forward and reverse primers and plasmids with the cloned TMPRSS2-ERG transcripts. The discrepancies in the amplicon sizes determined by RT-PCR and cloning/sequencing were due to the presence of additional sequences derived from the vector. The presence and the length of chimeric recombinant genes were determined by sequencing of the cloned fragments; (**c**) sequencing of the TMPRSS2-ERG transcripts at the fusion sites. TMPRSS2 and ERG sequences are shown in blue and yellow, respectively; (**d**) two new TMPRSS2-ERG fusion transcripts produced by the G > A substitution in the CGG triplet resulting in the formation of the CAG site; (**e**) identification of TMPRSS2-PPP6R1 recombinant transcript formed by interchromosomal translocation; PPP6R1 RNA ID, - NM_014931.

**Table 1 ijms-24-02830-t001:** Expression of AMACR, EZH2, and TMPRSS2-ERG in the cancerous and normal prostate tissues and early PCCs.

Summary of the AMACR, EZH2, TMPRSS2 Production in Cancer and Control Tissue Samples						
	Cancer Tissue			Normal Tissue		
Name of the marker	AMACR	EZH2	TMPRSS2-ERG	AMACR	EZH2	TMPRSS2-ERG
Number of the tested samples	18	18	18	15	15	15
Number of the positive tests	18 (100%)	18 (100%)	4 (22%)	1 (6.6%)	1 (6.6%)	0 (0%)
Frequency of the AMACR, EZH2, TMPRSS2 production in cancer tissue and corresponding early PCCs						
Number of the tested tissue samples	8	8	4	8	8	4
Number of the positive tissue tests	8 (100%)	8 (100%)	4 (100%)	1 (12.5%)	1 (12.5%)	0 (0%)
Number of the tested PCCs samples	8	8	4	8	8	ND
Number of the positive PCCs tests	8 (100%)	8 (100%)	4 (100 %)	0 (0%)	0 (0%)	

ND—not determined.

**Table 2 ijms-24-02830-t002:** Primers for amplification of cancer, epithelial, and mesenchymal markers in prostate tissues and derived PCCs.

Gene	Forward Primer, 5′-3′	Reverse Primer, 5′-3′	Amplicon Size, bp
*CK5*	TTCATCGACAAGGTGCGGT	TGAGGTGTCAGAGACATGCG	423
*TP63*	GTCCCAGAGCACACAGACAA	GAGGAGCCGTTCTGAATCTG	267
*CK18*	TGGTCACCACACAGTCTGCT	CCAAGGCATCACCAAGATTA	348
*AR1*	GACATGCGTTTGGAGACTGC	TTTCTTCAGCTTCCGGGCTC	294
*AMACR*	TGGCCACGATATCAACTATTTGG	ACTCAATTTCTGAGTTTTCCACAGAA	247
*EZH2*	TGCGACTGAGACAGCTCAAG	GCGCAATGAGCTCACAGAAG	164
*VIMENTIN*	GAGAACTTTGCCGTTGAAGC	GCTTCCTGTAGGTGGCAATC	163
*TMPRSS2-ERG*	CGCGAGCTAAGCAGGAG	GTCCATAGTCGCTGGAGGAG	180
*GAPDH*	CCATCTTCCAGGAGCGAGA	GGCAGTGATGGCATGGACTGT	326

## Data Availability

The data are available upon reasonable request to the corresponding author.

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
