# Peer review of "Early Cell Cultures from Prostate Cancer Tissue Express Tissue Specific Epithelial and Cancer Markers"

_ijms, 2023, doi:10.3390/ijms24032830_

Round 1
Reviewer 1 Report
Comments to manuscript ijms-2033155 The Early Cell Cultures from Prostate Cancer Tissue Produce Tissue Specific Epithelial and Cancer Markers
This manuscript describes in detail the procedures for carrying out primary cultures of prostate cells from normal and cancerous tissue, as well as the obtention of organoids and the characterisation of different culture stages through different markers linked to the epithelial-mesenchymal transition.
There are numerous publications of study models for prostate cancer, however, what is presented in the manuscript is one more approach that can be used for various studies as proposed. Unfortunately, marker-based characterisation is limited to a few genes, when it is the type of study that merits at least a discrete panel of expression markers either by array or by NGS. Finally, the description of the model falls somewhat short of what I believe could be improved and no substantial contribution to the field is evident. On the other hand, the writing can be a bit confusing and there are even data that are not presented where indicated, so I attach the pdf of the manuscript with some marks and comments. In particular the characterisation of translocations is very confusing and the finding of new isoforms is not completely linked to a specific phenotype. A careless product has been submitted, so I think it needs to be carefully restructured before resubmission.

Author Response
Dear reviewer # 1,
Thanks for reading and doing comments on our manuscript. All your notes and comments in the attached PDF file were used to correct the manuscript. In the revised version the changes outlined by red. We have tried to do our best and believe that you find the updated manuscript is satisfactory for publication.
Sincerely yours
Boris Popov

Reviewer 2 Report
In this paper, the authors investigate the abilities of the prostate primary 2D cell cultures from prostate cancer patients to produce epithelial and cancer markers, aiming to expand the early markers of genes for prostate cancer diagnostics. Population of prostate epithelial cells from cancer and normal tissue contains cancer or normal cancer tissue specific cells which in 3D culture in matrigel form patients derived organoids (PDO). They find that early prostate cancer contain tissue-specific stem cells which grow in 3D culture and form PDO similar to those from the prostate tissue. All type of PDO express the markers of prostate basal, luminal epithelial, while PDO from prostate cancer tissue and corresponding PCC produce cancer markers AMACR, TMPRSS2-ERG and EZH2. Generally, this paper has several shortcomings in both experimental design and result evaluation. My decision on this paper is reconsider after resubmission.
1. How do the authors selected the cancer marker candidates to be tested? In this paper, the authors mainly test the genes AMACR and TMPRSS2-ERG through RT–PCR, but the existing prostate cancer genes includes many genes: ACSL3, AR, AXIN1, BRAF, CANT1, CD209, DDX5, ELK4, ERG, ETV1, ETV4, ETV5, FOXA1, HERPUD1, HMGN2P46, HNRNPA2B1, KLF6, KLK2, NCOR2, NDRG1, PTEN, RAF1, SALL4, SLC45A3, SPOP, TMPRSS2, ZFHX3, ZNF479 (from well curated database Cancer Gene Census (CGC) [1]: https://cancer.sanger.ac.uk/census). In CGC, the gene TMPRSS2 is part of the TMPRSS2-ERG fusion, and the AMACR is not included since it is a candidate marker. Why the authors choose the two candidates instead of other potential genes?
[1] Sondka, Z., Bamford, S., Cole, C. G., Ward, S. A., Dunham, I., & Forbes, S. A. (2018). The COSMIC Cancer Gene Census: describing genetic dysfunction across all human cancers. Nature Reviews Cancer, 18(11), 696-705.
2. The authors do not analyze whether the two candidates are subgroup-specific markers for prostate cancer [2-3], although they have highlighted the heterogeneity of prostate cancer. Are the tested candidate genes associated with all samples of prostate cancer, or only a subset of samples? The analysis on cancer heterogeneity should also be conducted in this paper.
[2] Xi, J., Yuan, X., Wang, M., Li, A., Li, X., & Huang, Q. (2020). Inferring subgroup-specific driver genes from heterogeneous cancer samples via subspace learning with subgroup indication. Bioinformatics, 36(6), 1855-1863.
[3] Liang, C., Niu, L., Xiao, Z., Zheng, C., Shen, Y., Shi, Y., & Han, X. (2020). Whole-genome sequencing of prostate cancer reveals novel mutation-driven processes and molecular subgroups. Life sciences, 254, 117218.
3. To evaluate the difference between early prostate cancer cells from cancer and normal tissue, the authors do not perform any statistical significance test on their results, and no quantitively analysis such as p-values are adopted on their comparison results.
4. Literature missing in bibliography: there are several existing studies that also focus on organoid for prostate cancer epithelial cells [4-6].
[4] Van Hemelryk, A., Mout, L., Erkens-Schulze, S., French, P. J., van Weerden, W. M., & van Royen, M. E. (2021). Modeling prostate cancer treatment responses in the organoid era: 3D environment impacts drug testing. Biomolecules, 11(11), 1572.
[5] Park, J. W., Lee, J. K., Phillips, J. W., Huang, P., Cheng, D., Huang, J., & Witte, O. N. (2016). Prostate epithelial cell of origin determines cancer differentiation state in an organoid transformation assay. Proceedings of the National Academy of Sciences, 113(16), 4482-4487.
[6] Zhou, L., Zhang, C., Zhang, Y., & Shi, C. (2021). Application of organoid models in prostate cancer research. Frontiers in Oncology, 11, 736431.
Minor comments:
1. The abbreviation PSCC is not defined in neither abstract nor main manuscript. Is that another abbreviation for PCC?
2. Page 2, line 91, wrong line break between E and zh2.
3. Typesetting: Mismatch indents. For example, the space in indents are distinct between paragraph 2 and 3, page 2.
4. Typos and grammatical errors should be checked thoughtfully.
Author Response
- How do the authors selected the cancer marker candidates to be tested? In this paper, the authors mainly test the genes AMACR and TMPRregulateSS2-ERG through RT–PCR, but the existing prostate cancer genes includes many genes: ACSL3, AR, AXIN1, BRAF, CANT1, CD209, DDX5, ELK4, ERG, ETV1, ETV4, ETV5, FOXA1, HERPUD1, HMGN2P46, HNRNPA2B1, KLF6, KLK2, NCOR2, NDRG1, PTEN, RAF1, SALL4, SLC45A3, SPOP, TMPRSS2, ZFHX3, ZNF479 (from well curated database Cancer Gene Census (CGC) [1]: https://cancer.sanger.ac.uk/census). In CGC, the gene TMPRSS2 is part of the TMPRSS2-ERG fusion, and the AMACR is not included since it is a candidate marker. Why the authors choose the two candidates instead of other potential genes?
answer - three cancer marker genes were used in this study: AMACR, EZH2 and TMPRSS2-ERG. First two genes are well known cancer markers (Bracken et al., 2003; Lloyd et al., 2013; Kong et al., 2020; Park et al., 2021). Both AMACR and EZH2 are overproduced in various types of cancer and their overproduction is usually not induced by mutations. The hyperproduction of AMACR provides cancer tissue by an additional energy required for growth, while elevation of the level of EZH2 in cancer cells is usually a consequence of inactivation of the pRb-E2F1 signaling pathway regulating the level and function of EZH2 (Bracken et al., 2003). The TMPRSS2-ERG is a recombinant fusion gene which specifically forms in some patients with prostate cancer and serves as a specific prostate cancer marker (St John et al., 2012). These three genes may not be included in the CGC database because their hyperproduction in cancer is not associated with mutations.
- The authors do not analyze whether the two candidates are subgroup-specific markers for prostate cancer [2-3], although they have highlighted the heterogeneity of prostate cancer. Are the tested candidate genes associated with all samples of prostate cancer, or only a subset of samples? The analysis on cancer heterogeneity should also be conducted in this paper.
Answer – although EZH2 is a reliable cancer marker, its production in localized prostate cancer (PCa) is not well studied. Additionally, there is no candidate marker to distinguish the dormant forms of PCa with high risk for progression into aggressive disease. Our experiments showed that EZH2 and AMACR are hyperproduced in all probes of localized prostate cancer tested while TMPRSS2-ERG was detected only in 22% of cases. These results suggest that EZH2 is not perfect marker to differ the indolent forms of PCa with high risk of transformation into CRPC. However, quantitative evaluation of the level of EZH2 in localized PCa may help to understand the connection of its overproduction with risk of transformation of the primary PCa into aggressive disease.
- To evaluate the difference between early prostate cancer cells from cancer and normal tissue, the authors do not perform any statistical significance test on their results, and no quantitively analysis such as p-values are adopted on their comparison results.
Answer – we completely agree, statistical significance test is required to verify the results obtained. The goal of this work was to evaluate the 2D prostate cell culture (PCC) as a new affordable preclinical model to study pathogenesis of the localized PCa. Such evaluation was based on assessment of expression in PCaT and derived early PCC three cancer markers: AMACR, EZH2 and TMPRSS2-ERG. We found that AMACR and EZH2 are produced in all tissue cancer samples used (100%), while TMPRSS2-ERG only in 22% of cases. In normal prostate tissue AMACR, EZH2 were detected in 6.3% of cases, while TMPRSS2-ERG was not detected at all (0%). Comparison of PCaT and corresponding early PCC on expression of the same markers showed that AMACR and EZH2 were produced in 100% of both, the cancer probes and in the derived PCC (Table 1). Additionally, we performed the quantitative statistical analysis on the number and size of organoids derived from prostate tissue, early and late PCC. The most sensitive value among organoids characteristics used was the number of big size PDO which significantly and continuously decreased in early and late PCC in comparison with those in prostate tissue. These results are presented on the updated Fig. 3.
- Literature missing in bibliography: there are several existing studies that also focus on organoid for prostate cancer epithelial cells [4-6].
[4] Van Hemelryk, A., Mout, L., Erkens-Schulze, S., French, P. J., van Weerden, W. M., & van Royen, M. E. (2021). Modeling prostate cancer treatment responses in the organoid era: 3D environment impacts drug testing. Biomolecules, 11(11), 1572.
[5] Park, J. W., Lee, J. K., Phillips, J. W., Huang, P., Cheng, D., Huang, J., & Witte, O. N. (2016). Prostate epithelial cell of origin determines cancer differentiation state in an organoid transformation assay. Proceedings of the National Academy of Sciences, 113(16), 4482-4487.
[6] Zhou, L., Zhang, C., Zhang, Y., & Shi, C. (2021). Application of organoid models in prostate cancer research. Frontiers in Oncology, 11, 736431.
Answer – we agree with this note and included the recommended papers into the manuscript and list of references.
Minor comments:
- The abbreviation PSCC is not defined in neither abstract nor main manuscript. Is that another abbreviation for PCC?
- Page 2, line 91, wrong line break between E and zh2.
- Typesetting: Mismatch indents. For example, the space in indents are distinct between paragraph 2 and 3, page 2.
- Typos and grammatical errors should be checked thoughtfully.
Answer – all noted errors were corrected.
Sincerely yours
Boris Popov

Round 2
Reviewer 2 Report
1. There are still missing references already mentioned in last version of comments. Such as curated database Cancer Gene Census (CGC) [1], and subgroup-specific markers for prostate cancer [2-3].
[1] Sondka, Z., Bamford, S., Cole, C. G., Ward, S. A., Dunham, I., & Forbes, S. A. (2018). The COSMIC Cancer Gene Census: describing genetic dysfunction across all human cancers. Nature Reviews Cancer, 18(11), 696-705.
[2] Xi, J., Yuan, X., Wang, M., Li, A., Li, X., & Huang, Q. (2020). Inferring subgroup-specific driver genes from heterogeneous cancer samples via subspace learning with subgroup indication. Bioinformatics, 36(6), 1855-1863.
[3] Liang, C., Niu, L., Xiao, Z., Zheng, C., Shen, Y., Shi, Y., & Han, X. (2020). Whole-genome sequencing of prostate cancer reveals novel mutation-driven processes and molecular subgroups. Life sciences, 254, 117218.
2. There are still typos and grammatical errors in this version of paper. For example, in abstract, line 22, the word "contain" in "Early PCC contain tissue-specific stem cells" should be "contains".
Author Response
Dear reviewer,
we added the references that you recommended and did English editing.
Sincerely yours,
Boris Popov
